# Value Shaping: Bias Reduction in Bellman Error for Deep Reinforcement Learning

## Abstract

The Bellman error plays a crucial role as an objective function in deep reinforcement learning (DRL), serving as a proxy for the value error. However, this proxy relationship does not guarantee exact equivalence between the two, as the Bellman error inherently contains bias that can lead to unexpected optimization behavior. In this paper, we investigate the relationship between the value error and the Bellman error, and analyze why the Bellman error is not a reliable proxy due to its inherent bias. Leveraging the linear structure of the Bellman equation, we propose a value shaping method to compensate for this bias by adjusting the reward function—while ensuring that such modifications do not alter the optimal policy. In practice, we initialize two parallel Bellman iteration processes: one for estimating the bias and the other for updating the value function with minimal bias. Our method effectively learns a low-bias Q-function, making it broadly applicable and easily integrable into existing mainstream RL algorithms. Experimental results across multiple environments demonstrate that our approach improves RL efficiency, achieves superior performance, and holds promise as a fundamental technique in the field of reinforcement learning.

## 1 Introduction

Minimizing Bellman error is central to reinforcement learning (RL) algorithms Sutton & Barto (2018). RL optimizes decision-making by solving Markov Decision Processes (MDPs), with its underlying logic relying on the Bellman equation Bellman (1966). The Bellman equation defines how the state-action value function (Q-function) connects current decisions to long-term returns, making accurate estimation of the value function crucial. This process involves minimizing value estimation error Fujimoto et al. (2018), and its optimization directly impacts policy learning and improvement. Since the state-action value function serves as the foundation for policy optimization, effectively fitting Riedmiller (2005), approximating Sutton et al. (1999), updating Munos & Szepesvári (2008), and modeling Bellemare et al. (2017) it has become the core challenges in RL research.

Minimizing Bellman error is essentially minimizing value error Sutton (1988). Value error is defined based on the optimal state-value function and is generally not directly computable. When the value error reaches zero, it indicates that the value function has been perfectly fitted. With the advancement of neural networks, deep reinforcement learning has found increasingly broad applications, making the efficiency of value function fitting more critical. By improving the accuracy of Bellman error estimation Omura et al. (2024), optimizing the value function fitting process Patterson et al. (2022), and enhancing its efficiency, we can not only improve policy optimization but also increase the sample efficiency of RL algorithms.

The Bellman error serves as a proxy for the Value error Fujimoto et al. (2022). However, this proxy relationship does not guarantee equivalence between the two errors - it represents a trade-off. During optimization, the bias in the Bellman error is often ignored. When minimizing the Bellman error, the bias may change while the Value error remains constant. Due to both the linear nature of the Bellman equation and the way we evaluate Value error (using L2 or L1 norms), there can potentially exist multiple equivalent proxies for the same Value error (as illustrated with specific examples in Sec.4). Therefore, choosing an appropriate Bellman error and improving the value update process can help enhance the accuracy and efficiency of value function approximation.

Previous research has reshaped the Bellman error from the perspective of the reward function and Q-value rather than addressing the inherent issues of the Bellman error itself. The reward function and target Q-value are two crucial components of the Bellman Error. The Bellman Error can be reshaped by modifying the reward function, as demonstrated in various works on reward shaping Naik et al. (2024), reward smoothing Schulman et al. (2015), reward scaling Haarnoja et al. (2018a;b) and dense reward design Pathak et al. (2017); Osband et al. (2018); Burda et al. (2018). Research focusing on target Q-values has primarily investigated methods to modify Q-value to promote diversity Van Seijen et al. (2017); Sun et al. (2022a) or stability Ioffe (2015); Gallici et al. (2024); Fujimoto et al. (2024), as well as modeling their underlying distributions Bellemare et al. (2017); Dabney et al. (2018); Hessel et al. (2018); Cetin & Celiktutan (2023).

In this paper, we investigate the origins of Bellman error bias and methods for its reduction. The intuitive idea is to use **Monotonically** increasing linear **reward transformation** that preserve the optimal policy while linearly reducing the bias in Bellman error through the transformed rewards. We name this method MRT (Monotonic Reward Transformation). In terms of implementation, MRT maintains two parallel Bellman iteration processes: one with default settings to predict Bellman error bias, and another that incorporates the predicted bias as reward compensation into the target Q-values, enabling low-bias Q-value updates. These low-bias Q-values are then used to guide policy optimization. Notably, our method is plug-and-play, offering broad applicability across different settings. We apply the MRT method to several mainstream algorithms, and experiments across multiple environments demonstrate that MRT significantly improves sample efficiency and boosts algorithm performance.

## 2 RELATED WORK

Reward signal plays a critical role in determining the success or failure of the RL algorithm. Reward shaping (RS) has been extensively discussed in previous research, we introduce "Value Shaping" (VS), which shares a strong connection with reward shaping through the Bellman Equation and is similarly built upon the concept of preserving the optimal policy.

**Reward shaping**   In general, any method that modifies the reward signal in RL can be considered a form of reward shaping. The concept of reward shaping was first introduced in (Ng et al., 1999), which utilizes reward information to distinguish between states, enabling better policy learning while emphasizing that the optimal policy remains unchanged. In contrast, in sparse reward environments, additional intrinsic rewards are often designed (Chentanez et al., 2004) to achieve a similar goal. For instance, methods like RND (Pathak et al., 2017; Osband et al., 2018; Burda et al., 2018) add reward signals based on prior knowledge, allowing the policy to learn more effectively. Other works modify the way prior knowledge is utilized (Hu et al., 2020; Chen et al., 2022) to achieve improved performance. Some studies also focus on policy optimization by altering the reward signal to reduce the variance of the gradient, such as in GAE (Schulman et al., 2015).

**Value shaping**   also includes multiple research area as value is updated based on Bellman equation, some related to Bellman error reduction Kumar et al. (2019),and chain effect reduction Tang & Berseth (2024).Some related to represent learning, such as Dueling networkWang et al. (2016), BatchNorm Ioffe (2015); Bhatt et al. (2019),Spectral Normalization Bjorck et al. (2021) designed for value normalization, LayerNorm Network Gallici et al. (2024); Fujimoto et al. (2024). Some related to modeling related distribution, such as reward distribution Van Seijen et al. (2017); Sun et al. (2022b), value distribution Bellemare et al. (2017); Hessel et al. (2018), which leads to a series of research about distributional reinforcement learning, such as value decomposition Rashid et al. (2020), value quantile regression Dabney et al. (2018). While some work make a progress on variance reduction, such as reward Centering Naik et al. (2024) for value variance reduction, Advantage baseline Mnih (2016) for policy gradient variance reduction. There are also many works that utilize value shaping, such as reward scaling in the SAC Haarnoja et al. (2018a;b) paper.

## 3 BACKGROUND

### 3.1 DEEP REINFORCEMENT LEARNING

Reinforcement learning (RL) is an optimization framework for tasks of a sequential nature (Sutton & Barto, 2018). Typically, tasks are defined as a Markov decision process (MDP) ($\mathcal{S}$, $\mathcal{A}$, $r$, $p$, $d_0$, $\gamma$), where $\mathcal{S}$ is a finite state space, $\mathcal{A}$ is a finite action space, $r : \mathcal{S} \times \mathcal{A} \rightarrow \mathbb{R}$ is a bounded reward function (i.e., $|r(s,a)| \leq R_{\max}$ for some $R_{\max} < \infty$), $p(\cdot \mid s,a)$ denotes the transition probability distribution over next states given $(s,a)$, $d_0$ is the initial state distribution, and $\gamma \in [0,1)$ is the discount factor. Actions are selected according to a policy $\pi$. The performance of a policy is measured by its discounted return

$$J_r(\pi) = \mathbb{E}_\pi \left[ \sum_t^\infty \gamma^t r(s_t, a_t) \right]. \tag{1}$$

In reinforcement learning, the Bellman equations Bellman (1966) describe the relationship between the value of a state (or state-action pair) and the expected return from that state onward. Below is the expected Bellman operator $\mathcal{T}$.

$$\mathcal{T}Q^\pi(s,a) = \mathbb{E}_{s' \sim p, a' \sim \pi} \left[ r(s,a) + \gamma Q^\pi(s',a') \right], \tag{2}$$

which relates the value of the current state-action pair to an expectation over the next state-action pair. Given an approximate value function $Q$ (distinguished from the true value function $Q^\pi$ by dropping the $\pi$ superscript) of a target policy $\pi$, we denote the **Bellman error** $\epsilon(s,a)$:

$$\epsilon_Q(s,a) := Q(s,a) - \mathbb{E}_{s' \sim p, a' \sim \pi} \left[ r(s,a) + \gamma Q(s',a') \right]. \tag{3}$$

In practice, the Bellman error is approximated by **Temporal Difference** (TD)Sutton (1988) **error** $\delta(i)$, for a transition $i := (s,a,r,s')$, the TD error is,

$$\delta_Q(i) := Q(s,a) - (r(s,a) + \gamma Q(s',a')),$$

where $a' \sim \pi$. The relationship between the TD error and the Bellman error can be described as follows:

$$\epsilon_Q(s,a) = \mathbb{E}_{s',a'}[\delta_Q(i)].$$

In policy evaluation, the main objective of interest is a loss (such as the MSE or L1) over the **value error** $\Delta_Q(s,a)$, the distance of an approximate value function $Q$ to the true value function $Q^\pi$ of the target policy $\pi$:

$$\Delta_Q(s,a) := Q(s,a) - Q^\pi(s,a). \tag{4}$$

Value error is often unmeasurable, as the true value function $Q^\pi$ is unobtainable without the underlying MDP. While both the Bellman error and the value error are defined with respect to $Q$, for simplicity we drop the subscript when the error terms are not in reference to a specific value function. The relationship between the Bellman error and the value error has been explored in prior work Fujimoto et al. (2022), and it is formalized in the following:

**Proposition 3.1** (Value error as a function of Bellman error)**.** *For any state-action pair $(s,a) \in \mathcal{S} \times \mathcal{A}$, with state action distribution $d^\pi(s',a'|s,a) = \frac{1}{1-\gamma} \sum_{t=0}^\infty \gamma^t p^\pi((s,a) \rightarrow s', t) \pi(a'|s')$, the value error $\Delta_Q(s,a)$ can be defined as a function of the Bellman error $\epsilon_Q$*

$$\Delta_Q(s,a) = \frac{1}{1-\gamma} \mathbb{E}_{(s',a') \sim d^\pi(\cdot|s,a)}[\epsilon_Q(s',a')]. \tag{5}$$

This proposition explains the relationship between value error and Bellman error from the perspective of the state distribution.

### 3.2 LINEAR REWARD TRANSFORMATION

In a Markov Decision Process (MDP), a linear transformation of the reward function generally does not affect the optimal policy, depending on the specific form of the linear transformation.

Suppose the transformed reward function is given by

$$r'(s, a) = \alpha \cdot r(s, a) + \beta$$

where $\alpha$ and $\beta$ are constants, with $\alpha > 0$. a linear transformation of the form $r'(s, a) = \alpha \cdot r(s, a) + \beta$ with $\alpha > 0$ will not affect the choice of the optimal policy. However, if $\alpha < 0$ (i.e., the transformation involves a negative scaling factor), it will reverse the reward priorities, causing originally higher rewards to become lower, which would affect the optimal policy. Therefore, maintaining $\alpha > 0$ is essential. We also refer to linear transformations with $\alpha > 0$ as MRT(**M**onotonic increasing linear **R**eward **T**ransformation). This name provides better understanding and intuition.

## 4 MITIGATING THE BIAS OF BELLMAN ERROR

By appropriately altering the shape of the target domain, we can make the Q-function easier to learn. This means that we should pay attention to the shape of Bellman error. In the following content, to simplify expressions, the inputs of functions will be omitted when it's unnecessary. For example, the state-value function $Q(s, a)$ will be written as $Q$, the reward function $r(s, a)$ will be written as $r$, and the bias function $b(s, a)$ will be written as $b$.

**Problem statement.** The Bellman equation provides a recursive and iterative framework for value learning, offering a method to estimate long-term returns. However, when function approximation and temporal difference learning are introduced, any bias introduced during the value update not only persists but also propagates throughout the iterative process Fujimoto et al. (2022); Farahmand et al. (2010). This is because both value error and Bellman error are evaluated based on the relative absolute difference between two target values.

Although this bias may not necessarily prevent the optimal policy from converging, it can significantly influence the learning process of the value function. To illustrate this phenomenon, we borrow an example Fujimoto et al. (2022) that demonstrates the existence of such bias.

*Remark* 4.1 (**Bellman error hides bias**). Let $Q^\pi$ be the true value function for some MDP and policy $\pi$. We define the following approximate value functions:

$$Q_1 = Q^\pi + 1 \qquad \text{(bias is correlated)},$$
$$Q_2 = Q^\pi \pm 1 \qquad \text{(bias is uncorrelated)},$$

where $\pm 1$ denotes a random operator, taking the value $+1$ or $-1$ with equal probability. In both cases, following Eq. 4, the value error, measured by MSE or L1 loss will be 1 for any state-action pair. Following Eq. 2, expanding the expectation of next Q-value, the Bellman error of $Q_1$ will be $1 - \gamma \mathbb{E}[1] = 1 - \gamma$, while the Bellman error of $Q_2$ will be $|\pm 1 - \gamma \mathbb{E}[\pm 1]| = 1$. This means that when using Bellman error to estimate value error, measured with MSE or L1 loss, there exists bias in Bellman error.

*Remark* 4.2 (**Bellman iteration propagates bias**). Due to the fact that Bellman iteration updates and unfolds through bootstrapping, the biases introduced during this process cannot be eliminated by its own mechanism. Instead, these biases accumulate and propagate throughout the iterations Farahmand et al. (2010). The formal statement is as follows (omitting the reward function):

$$\mathcal{T}Q' = Q' + 1 \qquad \text{(next time step Q-value)},$$
$$\mathcal{T}Q = Q + 1 \qquad \text{(current time step Q-value)},$$
$$Q = \mathbb{E}[Q' + 1] + 1 \qquad \text{(bias is accumulated)}.$$

It can be seen that bias accumulates during the Bellman iteration. Bias can be introduced in various forms, and our main concern is how to eliminate or reduce its impact on value learning. Next, we discuss the sources of bias and explore methods to mitigate it.

### 4.1 ORIGINS OF BIAS AND REWARD COMPENSATION

From the previous discussion, it is clear that irrelevant biases cannot be controlled. If they follow a normal distribution, they also seem not to affect the estimation of values. Therefore, our focus should be on relevant biases, particularly those that are dependent on the state and action.

Regarding biases that depend on the state and action, we can draw the following conclusions:

**Proposition 4.3** (Bias arises from accumulated Bellman errors.)**.** *For any state-action pair* $(s_0, a_0) \in \mathcal{S} \times \mathcal{A}$, *the value error* $\Delta_Q(s_0, a_0)$ *is the accumulation of the Bellman errors* $\epsilon_Q$ *over future time steps:*

$$\Delta_Q(s_0, a_0) = \epsilon_Q(s_0, a_0) + \mathbb{E}_\pi \left[ \sum_{t=1}^{\infty} \gamma^t \epsilon_Q(s_t, a_t) \right]. \tag{6}$$

This proposition explains the relationship between value error and Bellman error from the perspective of trajectory interaction. When minimizing the value error, we focus on the Bellman error but neglect the Bellman errors from the remaining interactions, which are the primary sources of bias.

Due to the linear nature of the Bellman operator, there exists a complementary relationship between the reward and the bias of the Bellman error. To account for more general cases, we do not assume that $\mathbb{E}_\pi \left[ \sum_{t=1}^{\infty} \gamma^t \epsilon(s_t, a_t) \right]$ is known. Instead, the more general assumption is as follows:

**Assumption 4.4.** We assume that $\epsilon_Q$ contains some unpredictable bias $b$. The ideal Bellman error can then be expressed as:

$$\epsilon_Q^* = \epsilon_Q - b \tag{7}$$

Previous work focused on minimizing $\epsilon_Q$ rather than minimizing $\epsilon_Q^*$, which could result in optimization operations that do not actually reduce $\epsilon_Q^*$. Instead, this may cause the individual bias to keep changing, even though the expectation of the bias does not necessarily change. It is additionally noteworthy that $\epsilon_Q^* = \Delta_Q$. Based on two key observations: the Bellman expectation equation is linear, and a linear transformation of the reward function does not affect policy convergence, it can be inferred that the bias in the Bellman error can be compensated for by modifying the reward. We know that the reward can be linearly transformed without affecting the convergence of the optimal policy:

$$r' = \alpha \cdot r + \beta$$

Therefore, we can replace the reward function of $r$ to $r'$, and perform the following transformation on the Bellman error:

$$\epsilon_Q' = \epsilon_Q + r - r' \tag{8}$$

This essentially implies that $r - r'$ can serve as a compensation term to balance the bias as much as possible, thereby simplifying the fitting of the Q-value. And in practice , we can adjust the parameter of to make the $r - r'$ be close to $-b$, such we can have better proxy error $\epsilon_Q'$ for the value error to fitting the Q-function. Intuitively, at a certain stage of the learning process, when the Bellman error is fixed, our additional objective is to minimize both the Bellman error and the bias. For example:

$$\min_{r'} |\epsilon_Q' - \epsilon_Q^*| = \min_{r'} |-b - r + r'| \tag{9}$$

The issue here is that we cannot possibly know what the expected bias $b$ actually is.

## 4.2 Minimize the transformed Bellman error

Actually, there's a trade-off that could potentially help address this. Specifically, we can minimize the following:

$$\min_{r'} ||\epsilon_Q'| - |\epsilon_Q^*|| \leq \min_{r'} \epsilon_Q' + \min_{r'} |\epsilon_Q^*| \tag{10}$$

Eq. 10 shows that the optimal $\epsilon_Q^*$ no longer requires optimization and is independent of $r'$, so it can be omitted. As a result, we obtain a very concise objective function, what we need to do is transform reward function to minimize $\epsilon_Q'$. This (Eq. 8) transformed Bellman error can be regarded as a better proxy for value error minimization than naive Bellman error. Eq. 10 leads to $\epsilon_Q' = |\epsilon_Q^*|$ or $\epsilon_Q^*$. Since we do not know the sign of $\epsilon_Q^*$, and the Q-value is typically reported to be overestimated Fujimoto et al. (2018), we consider this relaxation (Eq. 10) acceptable by minimizing this objective periodically.

Now let's bring the problem back into the framework of reinforcement learning. Let's rethink the effect of the reward transformation, to derive some practice idea. Refer to Eq. 1, maximizing the sum of transformed rewards does guarantee a policy that also maximizes original rewards:

$$\arg\max_{\pi \in \Pi} J_{r'}(\pi) = \arg\max_{\pi \in \Pi} J_r(\pi).$$

In this formula, $r'$ is not fixed. Constantly changing $r'$ implies that collecting trajectories under different reward scales will result in data with varying reward scales. Training with data from different scales is almost impossible. Next, we will provide a solution to this problem, and explain why our method is value shaping rather than reward shaping.

### 4.3 VALUE SHAPING IS ALL YOU NEED

The reward shaping method evaluates the agent's performance based on the reshaped reward. If we consider our MRT method as a reward shaping approach, the resulting optimization objective is:

$$\max_{\pi' \in \Pi} J_{r'}(\pi')$$
$$\text{subject to } \ J_r(\pi') - \max_{\pi} J_r(\pi) = 0$$
$$r' = \arg\min \mathbb{E}[\epsilon_Q'^2].$$

Here, we consider minimizing all Bellman errors, so we use the MSE loss. It is worth noting that some reward shaping methods do not satisfy the second constraint and instead heuristically modify the reward signal. Based on the fact that our purpose in modifying the reward is not to enhance the reward function in a specific aspect, our objective can therefore simply be:

$$\max_{\pi \in \Pi} J_r(\pi)$$
$$\text{subject to } \ J_{r'}(\pi) - \max_{\pi'} J_{r'}(\pi') = 0$$
$$r' = \arg\min \mathbb{E}[\epsilon_Q'^2]. \tag{11}$$

Solving this optimization problem can be understood as proposing a policy $\pi'$ that maximizes $J_{r'}$ and then verifying whether the proposed $r'$ is feasible by improving the accuracy of the Bellman error. Referring to Eq. 9, we know that the optimal reward modification is given by $r'(s, a) = r(s, a) + b(s, a)$. This implies that we do not need to know the exact form of $r$. If we can accurately estimate $b$, then by using $\hat{r}' = r + \hat{b}$, we can complete the closed-loop implementation of our method. Although $b$ is unknown, we can reasonably infer that if we maintain an estimate $\hat{b}$, it should be as close as possible to $\epsilon_Q$ to minimize Eq. 7. Referring to Eq. 6, we know that the bias is related to the accumulated Bellman error. Referring to Eq. 11, the argmin of the sampled Bellman error is the expectation of the Bellman error, which implies that $\hat{b} = -\mathbb{E}[\epsilon_Q]$.

In practice, we initialize two value-learning processes. One follows the standard setting and is used to estimate the bias in the Bellman error, while the other, referred to as the lower-bias $Q$, is designed to learn a value function with reduced bias. By maintaining these two independent value-learning processes, our approach simultaneously mitigates value estimation errors and prevents error propagation. Referring to Eq. 2, we first initialize a standard Bellman iteration:

$$\mathcal{T}Q_1(s, a) = \mathbb{E}_{r,s'\sim p, a'\sim \pi}\left[r + \gamma Q_1(s', a')\right]. \tag{12}$$

The function $Q_1$ is updated to minimize the difference between $Q_1$ and $\mathcal{T}Q_1$. This process allows us to estimate the bias, denoted as $\hat{b}$, which is then used as the target for $Q_2$:

$$Q_1'(s, a) = \mathbb{E}_{r,s'\sim p, a'\sim \pi}\left[r + \hat{b} + \gamma Q_1(s', a')\right]. \tag{13}$$

The function $Q_2$ is then updated to minimize the difference between $Q_2$ and the low-bias target $Q_1'$. Following the delayed update trick Fujimoto et al. (2018), our framework periodically assigns $Q_2$ to $Q_1$ to achieve a low-variance Q-value update. Our method is considered a **value shaping** approach because the target Q-values are explicitly modified. Compared to reward centering Naik et al. (2024), our approach instead centers the Bellman error.

## 5 EXPERIMENTS

We conducted experiments on six continuous control tasks using the Mujoco Todorov et al. (2012) platform. The environments range from simple to complex, specifically: Swimmer-v4, Hopper-v4, HalfCheetah-v4, Ant-v4, HumanoidStandup-v4, and Humanoid-v4. The experiments were run on a hardware platform consisting of four Intel Gold 6230 CPUs and four RTX 3090 GPUs. Each algorithm was executed six times using random seeds from 1 to 6. Evaluated with 1M time steps, TD3 consumes 63.5 minutes, while TD3+MRT consumes 72.1 minutes. Practical algorithm and hyper-parameters are in the appendix( see Algorithm 1 and Table 2).

We compared three of the most well-known baseline algorithms in deep reinforcement learning, each of which has had a significant impact on the field. TD3 Fujimoto et al. (2018) mitigates the overestimation of predicted values and stabilizes the value function update process. RRS Sun et al. (2022a) shifts the reward, leading to a different initialization of the Q-function, which enhances performance by avoiding suboptimal solutions through diverse exploration. TD7 Fujimoto et al. (2024) modifies the neural network architecture and the inputs to the Q-function, achieving the strongest empirical performance.

### 5.1 THE IMPACT OF BIAS REDUCTION ON POLICY OPTIMIZATION

We first consider the impact of bias reduction on policy optimization. Ideally, bias reduction can improve the accuracy of the Q-function, thereby having a positive effect on on policy optimization. The purpose of bias reduction is also to accelerate policy optimization during the learning process. An inaccurate Q-function leads to inaccurate policy gradients. Therefore, evaluating the impact of bias reduction from the perspective of policy optimization can indirectly reflect its effectiveness. Based on this, we assess the effect of MRT on the three baseline algorithms. The results are shown in Fig. 1.

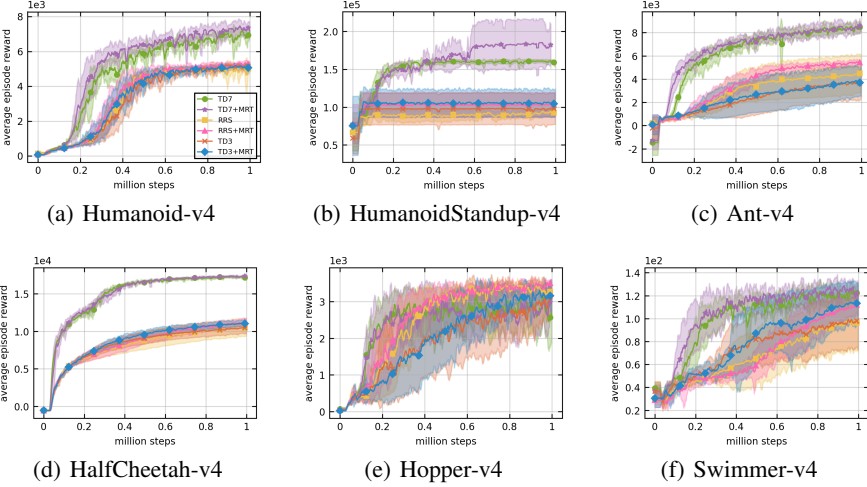

(a) Humanoid-v4     (b) HumanoidStandup-v4     (c) Ant-v4

(d) HalfCheetah-v4     (e) Hopper-v4     (f) Swimmer-v4

Figure 1: Learning curves of all algorithms. The x-axis represents time steps, with one million interaction steps, and the y-axis represents the average episode return.

From the results shown in the figure, we can observe that in more complex environments, such as those involving the control of two humanoid robots (Fig. 1(a-b)), MRT demonstrates a significant advantage. By providing more accurate Q-value estimations, MRT enhances the efficiency of policy optimization. It not only improves the sample efficiency of TD7 but also outperforms both RRS and TD3. In the Humanoid-v4 environment, MRT enhances the final convergence performance of the original algorithm. In the Ant-v4 environment, MRT provides noticeable benefits to the baseline algorithms in the early stages of training. However, in the HalfCheetah-v4 environment, the difference is less pronounced. Since this environment has the second-largest reward scale (with HumanoidStandup-v4 being the largest), the stability of the algorithm may play a role in this observation. Indeed, in both of these environments, the learning process appears relatively stable. In

the Hopper-v4 environment, the TD7 algorithm, which incorporates LayerNorm, does not seem to handle the task well. MRT does not significantly improve TD7's performance in this case but instead provides more noticeable benefits to the RRS algorithm, especially in the early training phase. Finally, in the Swimmer-v4 environment, MRT enhances learning accuracy by reducing bias, leading to improved sample efficiency across all three algorithms.

## 5.2 THE IMPACT OF BIAS REDUCTION ON VALUE UPDATE

After verifying the impact of reducing Bellman error bias on policy optimization, the next focus is on its effect on value updates. The value function heavily depends on the magnitude and accuracy of the Bellman error. While bias reduction has an indirect impact on policy optimization, it directly influences value updates. Since Q-function optimization is achieved by minimizing the Bellman error, the two are closely related. Analyzing Q-value trends provides insights into the entire training process. Therefore, we recorded the estimated Q-values based on the sampled transitions during training. Typically, as the policy improves, the Q-value increases accordingly. However, different algorithms affect the Q-value update process differently, reflecting their influence on value estimation. The evaluation results of the Q-values are shown in Figure 2.

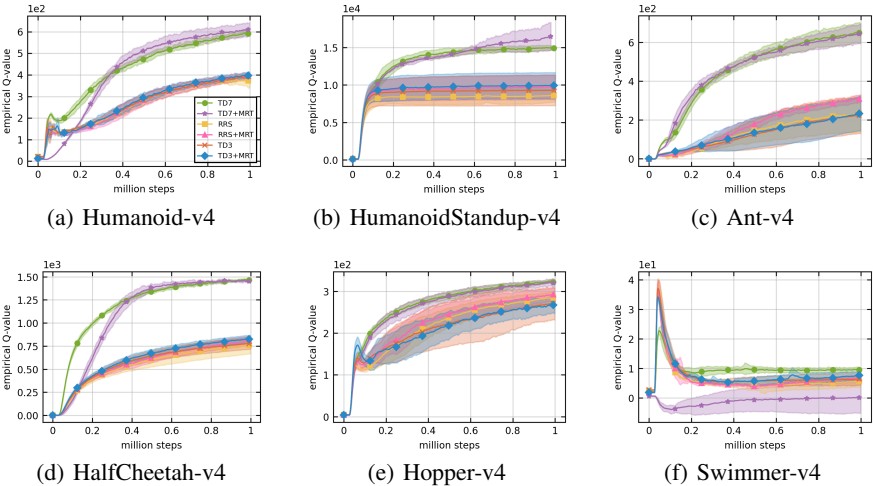

Figure 2: Empirical Q-value of all algorithms. The x-axis represents time steps, with one million interaction steps, and the y-axis represents the estimated empirical Q-value over the sampled transitions.

From Figure 2, we can observe that, compared to baseline algorithms, the Q-values in the early training phase are generally lower when using the MRT algorithm. This is evident in environments such as Humanoid-v4, HumanoidStandup-v4, HalfCheetah-v4, and Swimmer-v4. The reason for this is that MRT reduces the original Bellman error, leading to smaller update magnitudes, which in some cases also results in higher accuracy. In the later training stages, if previous algorithms were limited by Q-value accuracy issues, our method's Q-values tend to catch up and even surpass them over time.

Another noteworthy observation is that, unlike policy evaluation results, Q-value updates are relatively stable. Although the collected samples represent only a subset of all possible transitions, Q-values generally continue to grow in most environments. However, in the Swimmer-v4 environment, due to the TD7 algorithm using priority sampling based on TD error and our method reducing the error magnitude, the Q-value update curve appears less consistent.

## 5.3 THE IMPACT OF BIAS REDUCTION ON BELLMAN ERROR

Besides the two previously mentioned metrics, the most important one we should focus on is the change in Bellman error, as our entire paper revolves around discussing it. This metric typically reflects the smoothness of the learning process, where a smaller Bellman error indicates convergence.

At the same time, a larger Bellman error suggests greater prediction errors, which can lead to instability. However, on the other hand, a larger Bellman error also implies a greater optimization step. Since it can serve both as an optimization objective and an evaluation metric, Bellman error has a dual nature. We aim to compare the average Bellman error across different algorithms to assess the potential impact of Bias Reduction on the results. The variation in Bellman error during the experiments is recorded in Fig. 3.

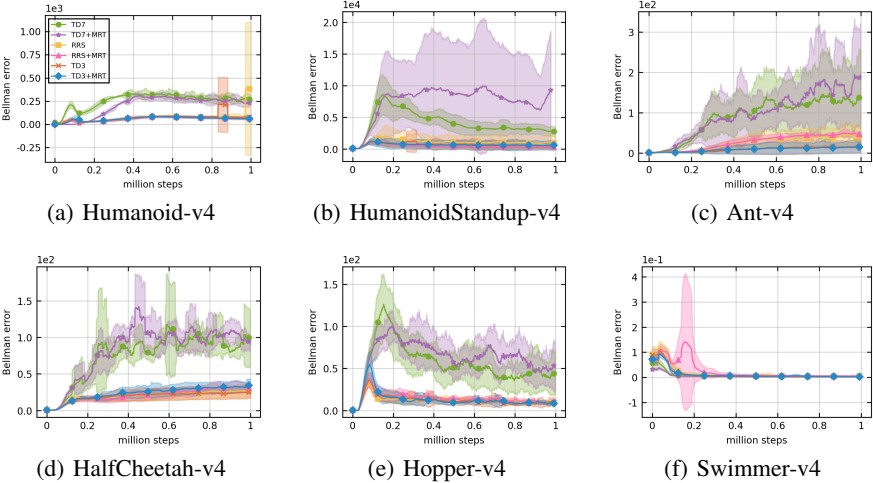

Figure 3: Empirical estimated square Bellman error of all algorithms. The x-axis represents time steps, with one million interaction steps, and the y-axis represents the average estimated square Bellman error over the sampled transitions.

Observing Fig. 3, we find that in the Humanoid-v4 environment, the MRT algorithm effectively avoids the late-stage anomalies in Bellman error compared to TD3 and RRS, while making the optimization process more stable when compared to TD7. In the HumanoidStandup-v4 environment, as the algorithms are still improving, our results have not yet fully converged. However, our method demonstrates greater stability compared to the RRS algorithm. Similarly, in the Ant-v4 environment, the MRT algorithm consistently exhibits a larger Bellman error than the TD7 algorithm. This is because TD7 employs priority sampling based on TD error and clips small TD errors, a technique that does not benefit our algorithm. Similar trends can be observed in the HalfCheetah-v4 and Hopper-v4 environments. Since both RRS and TD3 use random sampling, the comparison between RRS and TD3 is more convincing. In the Swimmer-v4 environment, the RRS+MRT algorithm shows a noticeable spike in Bellman error, because that a larger Bellman error results in larger step sizes, making it more effective in environments requiring exploration.

## 6 CONCLUSION

This paper investigates the reduction of Bellman error bias through linear reward transformation. By leveraging the fact that linear reward transformations do not affect policy convergence, we estimate the bias in the Bellman error and incorporate it into the reward function to influence the value update process. This process is carried out using two parallel Bellman iterations, where bias estimation techniques and linear reward transformation are employed. This simplifies the MRT algorithm, making it applicable to any deep reinforcement learning algorithm. Experimental results show that reducing Bellman error bias improves sample efficiency. Given the critical role of Bellman error in reinforcement learning, there is significant potential for further research. Future work will focus on developing more advanced techniques for bias prediction and reduction.

ETHICS STATEMENT

This research did not involve human participants, personal data, or animals, and therefore did not require institutional ethics approval. All experiments were conducted using publicly available datasets and simulated environments, ensuring that no privacy or safety concerns arise.

REPRODUCIBILITY STATEMENT

We provide the implementation code and configuration files in the supplementary material. All reported results are averaged over six random seeds (1, 2, 3, 4, 5, 6). Shaded regions in the figures denote one standard deviation around the mean. Details of the hardware platform and computational time are presented at the beginning of the Experiments section.

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

## A    PROOF

In the proof section, for better readability, we have simplified some expressions. For example, we use $\Delta$ to represent $\Delta_Q$.

### A.1    PROOF OF PROPOSITION 3.1

**Proposition 3.1 (Value error as a function of Bellman error).** *For any state-action pair* $(s, a) \in \mathcal{S} \times \mathcal{A}$, *with state action distribution* $d^\pi(s', a'|s, a) = \frac{1}{1-\gamma} \sum_{t=0}^{\infty} \gamma^t p^\pi((s, a) \to s', t) \pi(a'|s')$, *the value error* $\Delta_Q(s, a)$ *can be defined as a function of the Bellman error* $\epsilon_Q$

$$\Delta_Q(s, a) = \frac{1}{1-\gamma} \mathbb{E}_{(s', a') \sim d^\pi(\cdot|s, a)} [\epsilon_Q(s', a')]. \tag{14}$$

*Proof.* We begin by stating results from Kakade & Langford (2002); Schulman (2015); Queeney et al. (2021). A policy $\pi$ induces a normalized discounted state visitation distribution $d^\pi$, where $d^\pi(s'|s, a) = \frac{1}{1-\gamma} \sum_{t=0}^{\infty} \gamma^t p^\pi((s, a) \to s', t)$. We write the corresponding normalized discounted state-action visitation distribution as $d^\pi(s', a'|s, a) = d^\pi(s'|s, a)\pi(a' \mid s')$, where we make it clear from the context whether $d^\pi$ refers to a distribution over states or state-action pairs.

First by definition, for state $s_1$ and action $a_1$, we have:

$$\mathbb{E}_{d^\pi}[\epsilon(s_1, a_1)] \tag{15}$$

$$= \sum_{s_1} d^\pi(s_1) \sum_{a_1} \pi(a_1 \mid s_1)\epsilon(s_1, a_1). \tag{16}$$

$$= (1-\gamma)\mathbb{E}_\pi \left[ \sum_{t=0}^{\infty} \gamma^t (Q(s_t, a_t) - r(s_t, a_t) - \gamma Q(s_{t+1}, a_{t+1})) \right] \tag{17}$$

$$= (1-\gamma)\mathbb{E}_\pi \left[ Q(s_0, a_0) - \sum_{t=0}^{\infty} \gamma^t r(s_t, a_t) \right] \tag{18}$$

$$= (1-\gamma)(Q(s_0, a_0) - Q^\pi(s_0, a_0)) \tag{19}$$

Then, we can derive similar result when we have state $s_{k+1}$ and action $a_{k+1}$,

$$\mathbb{E}_{d^\pi}[\epsilon(s_{k+1}, a_{k+1})] = (1-\gamma)(Q(s_k, a_k) - Q^\pi(s_k, a_k)) \tag{20}$$

Finally, we have that:

$$\frac{1}{1-\gamma} \mathbb{E}_{(s', a') \sim d^\pi(\cdot|s, a)}[\epsilon(s', a')] = Q(s, a) - Q^\pi(s, a) = \Delta_Q(s, a)$$

$\square$

### A.2    PROOF OF PROPOSITION 4.3

**Proposition 4.3 (Bias stems from accumulated Bellman errors.)** *For any state-action pair* $(s_0, a_0) \in \mathcal{S} \times \mathcal{A}$, *the value error* $\Delta_Q(s_0, a_0)$ *is the accumulation of the Bellman errors* $\epsilon_Q$ *over future time steps:*

$$\Delta_Q(s_0, a_0) = \epsilon_Q(s_0, a_0) + \mathbb{E}_\pi \left[ \sum_{t=1}^{\infty} \gamma^t \epsilon(s_t, a_t) \right] \tag{21}$$

*Proof.* First by definition:

$$\Delta(s, a) := Q(s, a) - Q^\pi(s, a) \tag{22}$$

$$\Rightarrow Q^\pi(s, a) = Q(s, a) - \Delta(s, a). \tag{23}$$

Then we can decompose value error:

$$\Delta(s,a) = Q(s,a) - Q^\pi(s,a) \tag{24}$$

$$= Q(s,a) - (r(s,a) + \gamma\mathbb{E}_\pi[Q^\pi(s',a')]) \tag{25}$$

$$= Q(s,a) - (r(s,a) + \gamma\mathbb{E}_\pi[Q(s',a') - \Delta(s',a')]) \tag{26}$$

$$= Q(s,a) - (r(s,a) + \gamma\mathbb{E}_\pi[Q(s',a')]) + \gamma\mathbb{E}_\pi[\Delta(s',a')] \tag{27}$$

$$= \epsilon(s,a) + \gamma\mathbb{E}_\pi[\Delta(s',a')] \tag{28}$$

$$\vdots \tag{29}$$

$$= \epsilon(s,a) + \gamma\mathbb{E}_\pi[\epsilon(s',a')] + \gamma^2\mathbb{E}_\pi[\Delta(s'',a'')]. \tag{30}$$

Finally, we derive:

$$\Delta_Q(s_0,a_0) = \epsilon_Q(s_0,a_0) + \mathbb{E}_\pi\left[\sum_{t=1}^\infty \gamma^t \epsilon(s_t,a_t)\right].$$

$\square$

## B   ALGORITHM

MRT initialize two Bellman iteration process, One is used for predict the bias of Bellman error, and the other one is used for learning an accurate value function with bias reduction from the Bellman error. Our method can seamlessly integrate with any DRL algorithm. In practice, after predicting the bias, we allocate a certain number of time steps for the $Q_2$ function in Eq. 13 to learn. After these time steps, we synchronize the parameters of $Q_2$ with $Q_1$ and the target network.

---

**Algorithm 1** Monotonic increasing linear Reward Transformation (MRT).

---

**Require:** $\theta, \bar{\theta}, \phi$, Replay Buffer $D$     ▷ Initial parameters $\theta, \bar{\theta}$ of the $Q$ function and $\phi$ of the target policy $\pi_\phi$.

1:   $\breve{\theta} \leftarrow \theta, \mathcal{D} \leftarrow \emptyset$                      ▷ Initialize parameters $\breve{\theta}$ of target Q-network
2:   **for** each iteration **do**
3:      **for** each environment step **do**
4:         Run policy $\phi$ in environment to collect transitions
5:         Store transitions into Buffer $D$
6:      **end for**
7:      **for** each training step **do**
8:         sample batch transition $(s,a,r,s')$ from Buffer
9:         update policy $\phi$ according to any DRL algorithm
10:       for each transition, compute the TD error
11:       update $\theta$ by minimizing the batch TD error
12:       estimated the bias with the TD error periodically
13:       update $\bar{\theta}$ with bias reduction target Q-value
14:      **end for**
15:   $\breve{\theta} \leftarrow \tau\bar{\theta} + (1-\tau)\breve{\theta}, \theta \leftarrow \tau\bar{\theta} + (1-\tau)\theta$
16: **end for**

**Ensure:** $\phi$                                              ▷ Optimized policy

---

In this paper, we do not specifically discuss the initialization of the Q-function or policy, as these steps have already been addressed in previous studies Haarnoja et al. (2018a). To apply our method to any DRL algorithm, the relative changes are as follows: in line 12, we estimate the bias of the Bellman error; in line 13, we update the low-variance Q-function; and in line 15, we adopt delayed updates to synchronize the parameters of the low-variance Q-function with those of other Q-functions. To maintain logical clarity, we simplify the representation of the target Q-function. The target Q-function may have multiple forms, as seen in the TD3 Fujimoto et al. (2018) paper. We clarify this detail here.

## C  DETAIL NUMERICAL RESULT

Some results in the figure are not very clear due to differences in data scales and overlapping curves, making comparisons less obvious. To provide a clearer analysis, we have reorganized all the results into a table, recording outputs every 0.2 million steps. The detailed results are shown in Table 1. The data in the table represent the mean of six results, along with one standard deviation, covering a 95% confidence interval. The best results are highlighted in bold.

Table 1: Numerical Result of Average Episodic Reward. (A) Humanoid-v4 (B) HumanoidStandup-v4 (c) Ant-v4 (D) HalfCheetah-v4 (E) Hopper-v4 (F) Swimmer-v4

| Env | Algo | 0.2M | 0.4M | 0.6M | 0.8M | 1M |
|-----|------|------|------|------|------|-----|
| (A) | TD7 | **2967.72 ± 1150.24** | 4822.53 ± 2116.07 | 6513.07 ± 340.57 | 6648.29 ± 713.04 | 6714.27 ± 619.08 |
|  | TD7+MRT | 2688.12 ± 1653.01 | **6053.62 ± 964.87** | **6831.41 ± 419.09** | **7124.76 ± 438.29** | **6902.72 ± 1663.1** |
|  | RRS | 737.9 ± 201.42 | 3268.5 ± 1601.4 | 5065.71 ± 146.34 | 5091.56 ± 173.47 | 5183.52 ± 171.04 |
|  | RRS+MRT | 782.18 ± 221.26 | 4277.43 ± 1115.75 | 4893.84 ± 532.89 | 4827.64 ± 1012.94 | 5132.51 ± 560.97 |
|  | TD3 | 875.45 ± 405.8 | 2596.93 ± 1302.98 | 4517.56 ± 704.81 | 4925.12 ± 271.16 | 5158.17 ± 245.98 |
|  | TD3+MRT | 677.92 ± 39.98 | 3186.55 ± 1645.28 | 4923.43 ± 221.93 | 4879.45 ± 309.63 | 5263.84 ± 187.92 |
| (B) | TD7 | 151725.22 ± 6682.32 | 159877.79 ± 2716.19 | 159979.43 ± 3166.79 | 160650.85 ± 1843.46 | 161599.67 ± 2129.93 |
|  | TD7+MRT | **155146.12 ± 10903.61** | **160334.25 ± 9870.83** | **182681.5 ± 32419.07** | **177913.47 ± 24701.51** | **172663.08 ± 15063.59** |
|  | RRS | 88700.35 ± 9331.45 | 89977.85 ± 9127.66 | 88891.87 ± 8366.44 | 91862.05 ± 5984.5 | 93510.03 ± 5916.96 |
|  | RRS+MRT | 102863.1 ± 16483.1 | 102950.0 ± 16465.94 | 102764.39 ± 16514.43 | 102804.6 ± 16129.92 | 103059.35 ± 16439.05 |
|  | TD3 | 97760.89 ± 21166.12 | 97894.46 ± 21151.11 | 97995.46 ± 20865.68 | 98024.74 ± 21252.07 | 98259.45 ± 20689.53 |
|  | TD3+MRT | 104253.44 ± 17759.69 | 103224.3 ± 17674.67 | 104419.95 ± 17800.85 | 105520.74 ± 18392.09 | 105509.46 ± 18378.86 |
| (C) | TD7 | 5066.0 ± 1385.38 | 6832.59 ± 708.66 | **8255.37 ± 622.25** | 7819.47 ± 1099.33 | 8079.26 ± 933.06 |
|  | TD7+MRT | **5999.98 ± 511.31** | **7214.9 ± 198.13** | 7893.69 ± 354.34 | **8217.03 ± 726.93** | **8781.44 ± 694.34** |
|  | RRS | 1161.07 ± 395.87 | 3107.51 ± 1207.38 | 3895.63 ± 1632.36 | 4318.46 ± 1722.3 | 4467.62 ± 1751.35 |
|  | RRS+MRT | 1386.77 ± 843.81 | 3547.72 ± 1141.62 | 4994.54 ± 189.86 | 4678.93 ± 1113.55 | 5529.2 ± 201.36 |
|  | TD3 | 1136.5 ± 373.53 | 1963.78 ± 1275.9 | 2884.04 ± 1501.95 | 3342.98 ± 1537.06 | 3734.29 ± 1478.07 |
|  | TD3+MRT | 1046.96 ± 274.71 | 2433.53 ± 1508.08 | 2771.54 ± 1600.17 | 3433.58 ± 1499.59 | 3972.38 ± 1357.83 |
| (D) | TD7 | 12461.65 ± 1131.17 | **16104.55 ± 593.02** | 16720.68 ± 208.9 | 17244.15 ± 182.15 | **17291.41 ± 200.61** |
|  | TD7+MRT | **12995.64 ± 633.1** | 16036.93 ± 761.71 | **16958.32 ± 307.61** | **17387.21 ± 258.62** | 17267.75 ± 215.09 |
|  | RRS | 6432.43 ± 570.43 | 8148.57 ± 1190.63 | 9172.71 ± 1223.24 | 10198.72 ± 1198.15 | 10551.27 ± 1215.57 |
|  | RRS+MRT | 6465.94 ± 426.35 | 8209.68 ± 1096.54 | 9359.76 ± 1140.33 | 10678.88 ± 655.21 | 11255.05 ± 593.87 |
|  | TD3 | 6585.94 ± 481.08 | 8670.89 ± 805.42 | 9582.96 ± 769.87 | 10054.63 ± 764.89 | 10590.11 ± 723.63 |
|  | TD3+MRT | 6789.49 ± 356.01 | 9055.14 ± 522.3 | 10188.93 ± 543.87 | 10186.0 ± 987.97 | 11179.5 ± 491.82 |
| (E) | TD7 | **2529.54 ± 843.66** | 2379.75 ± 884.58 | 3189.78 ± 575.91 | 3286.19 ± 800.75 | 2410.19 ± 961.2 |
|  | TD7+MRT | 2445.37 ± 1162.01 | 2509.6 ± 945.57 | 2571.26 ± 747.59 | 2812.32 ± 749.38 | 3076.99 ± 940.72 |
|  | RRS | 1151.63 ± 761.39 | 2505.71 ± 983.89 | **3375.09 ± 107.06** | 3400.35 ± 153.31 | 2691.38 ± 1147.2 |
|  | RRS+MRT | 1741.77 ± 1120.7 | **3338.48 ± 165.87** | 3251.63 ± 344.11 | **3494.09 ± 81.85** | **3537.2 ± 56.93** |
|  | TD3 | 1331.36 ± 1069.89 | 1962.45 ± 1203.24 | 2545.1 ± 1201.5 | 2025.49 ± 993.43 | 3118.38 ± 518.57 |
|  | TD3+MRT | 890.19 ± 762.45 | 2014.25 ± 1052.76 | 2394.97 ± 1033.55 | 2815.3 ± 850.2 | 3378.25 ± 105.71 |
| (F) | TD7 | 70.94 ± 23.5 | 104.81 ± 12.73 | **118.34 ± 5.57** | **123.28 ± 12.0** | 114.36 ± 16.28 |
|  | TD7+MRT | **99.95 ± 24.68** | 107.13 ± 31.74 | 95.76 ± 32.4 | 118.33 ± 24.88 | **128.21 ± 8.79** |
|  | RRS | 44.36 ± 4.58 | 58.5 ± 19.7 | 73.61 ± 22.92 | 88.23 ± 25.78 | 99.72 ± 31.57 |
|  | RRS+MRT | 48.2 ± 7.84 | 48.59 ± 6.51 | 76.83 ± 15.84 | 100.01 ± 18.17 | 112.91 ± 10.89 |
|  | TD3 | 50.9 ± 6.33 | 75.05 ± 22.54 | 90.11 ± 21.14 | 98.72 ± 24.15 | 96.13 ± 25.5 |
|  | TD3+MRT | 50.78 ± 9.35 | 64.27 ± 45.49 | 98.64 ± 25.56 | 98.18 ± 28.36 | 112.67 ± 12.8 |

From the table, we can observe that the training stability of TD7 is not ideal in certain environments. For example, the final converged result is sometimes worse than the maximum value achieved during training, which affects the stability of our method on TD7 as well. The fundamental reason for this issue lies in the TD7 algorithm's weighted sampling of experience replay based on TD error, without considering the impact of weighting on convergence. This can be seen from the data at different training stages. For instance, in the HumanoidStandup-v4 environment, the best result for the TD7+MRT algorithm appears at 0.6M time steps. Similarly, in the Ant-v4 environment, TD7 achieves its best result at 0.6M time steps.

## D  HYPER-PARAMETERS

Taking the TD3 algorithm as an example, MRT introduces only one additional hyperparameter—the period for estimating the bias.

Table 2: Hyper-parameters

| Parameter | Value |
|---|---|
| *Shared (TD3)* | |
| optimizer | Adam |
| learning rate | $3 \cdot 10^{-4}$ |
| discount ($\gamma$) | 0.99 |
| replay buffer size | $10^6$ |
| number of hidden layers (all networks) | 2 |
| number of hidden units per layer | 256 |
| number of samples per minibatch | 256 |
| nonlinearity | ReLU |
| target smoothing coefficient ($\tau$) | 0.005 |
| target update interval | 1 |
| gradient steps | 1 |
| *MRT* | |
| estimated bias update interval | 250 |

## E  MORE EXPERIMENTS

### E.1  THE IMPACT OF BIAS REDUCTION ON EXPLORATION

Exploration is also crucial in reinforcement learning. While bias reduction serves as an optimization technique, examining its impact on exploration provides indirect insight into how bias reduction affects policy learning. In most cases, a good policy naturally leads to effective exploration, although effective exploration does not necessarily guarantee stable convergence.

We have recorded the results of exploration, as shown in Fig. 4.

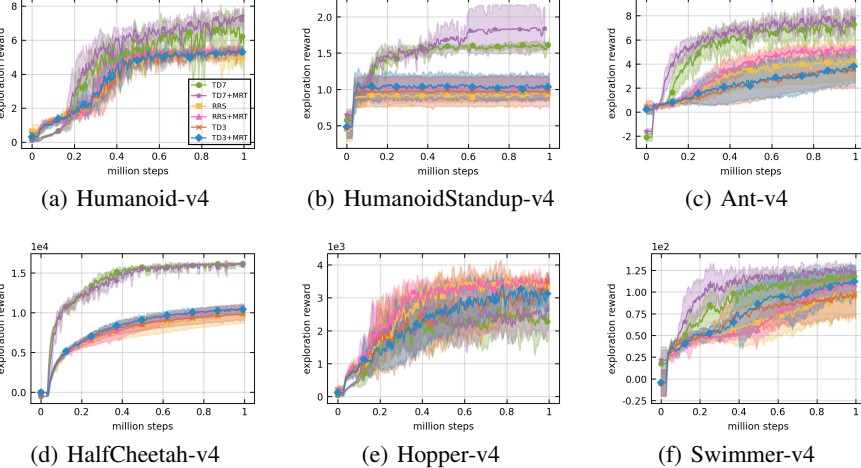

(a) Humanoid-v4          (b) HumanoidStandup-v4          (c) Ant-v4

(d) HalfCheetah-v4          (e) Hopper-v4          (f) Swimmer-v4

Figure 4: Exploration reward of all algorithms. The x-axis represents time steps, with one million interaction steps, and the y-axis represents the average episodic exploration reward.

From Fig. 4, we observe that after applying the MRT method, the exploration performance is almost consistently better than that of the baseline algorithms. This aligns with the policy evaluation results, particularly in the Humanoid-v4, HumanoidStandup-v4, and Ant-v4 environments, where the exploration performance continues to improve. This also explains why Bellman error contin-

ues to increase in the later stages of training. Bias reduction enables a smoother learning process for Q-values, reducing the time required for value fitting and allowing additional opportunities for exploration. This additional exploration leads to better policies, which in turn enhance the value of earlier states and increase Q-value errors—ultimately resulting in improved performance.

## F    LIMITATION

In terms of computational efficiency, our method introduces an additional Q-function update, which can potentially increase computational cost. Although compared to prior methods such as the RRS algorithm, our approach uses fewer Q-functions overall, this added update still contributes to higher computation overhead. Specifically, if we consider the cost of Q-function computation alone, and take the original DQN algorithm as a baseline, DQN only computes a single target Q-function and updates the current Q-function once per step. TD3 computes two target Q-functions and updates the current Q-function once. TD3+MRT adds one more Q-function update on top of TD3. In our method, while we maintain a relatively efficient structure, the additional Q-function update introduces a similar level of computational cost. Furthermore, we peridocally approximate the bias of the Bellman error in our method. However, this estimation may be inaccurate in some situations. We assume that Q-values are overestimated, but the extent of this overestimation and the frequency at which it occurs are difficult to determine precisely.

