# OpenReview forum: "Value Shaping: Bias Reduction in Bellman Error for Deep Reinforcement Learning"
_ICLR.cc/2026/Conference — ICLR 2026 Conference Withdrawn Submission_

### Official Review · Reviewer_M7qB · 2025-10-21

**Soundness:** 1
**Presentation:** 1
**Contribution:** 1
**Rating:** 2
**Confidence:** 3

**Summary:**

This paper presents a reinforcement learning algorithm that linearly transforms the reward function to improve sample efficiency.
Specifically, the authors propose a method to compensate for the bias in the target value of the Q-function.
In the proposed approach, two functions are used to approximate the Q-function.
If my understanding is correct, the bias in the value function is estimated as the mean of the Bellman error computed from one of the approximated Q-functions, and this estimated bias is then used to modify the target Q-value for the other approximated Q-function.
The proposed method was evaluated on benchmark tasks in Gymnasium with MuJoCo environments.

**Strengths:**

- The idea of transforming the reward function is an interesting research direction for improving the sample efficiency of RL algorithms.

- The proposed method is agnostic to specific policy update algorithms and can be combined with various RL methods.

- The approach appears simple and easy to implement, provided that the necessary details are clearly described.

**Weaknesses:**

- The presentation is not well organized and is difficult to follow in some parts.

- The performance improvement appears limited.

Although the paper discusses the difference between the value error and the Bellman error, it is unclear how the proposed method addresses the challenge of estimating the value error. In lines 236–237, the paper states: "It is additionally noteworthy that  $\epsilon^*_{Q}=\Delta_{Q}$."

However, the definition of $\epsilon^*_Q$ seems to be missing, and it is unclear how this relationship is mathematically justified.

Moreover, the description of the proposed algorithm is ambiguous.
If I understand correctly, the bias in the value function is estimated as the mean of the Bellman error,
$\hat{b} = r_i + \gamma Q_1 (s'_i, a'_i) - Q_1 (s_i, a_i)$,
and the target value for $Q_2$ is computed as
$r_i + \gamma Q_2 (s'_i, a'_i) + \hat{b}$.
However, I am not entirely sure if this interpretation is accurate.
If it is, I do not think the proposed algorithm can correctly reduce the bias, since the bias in $Q_1$ differs from that in $Q_2$, and the Bellman error is generally not equivalent to the value error.
The authors should revise the manuscript to clarify the proposed algorithm and its theoretical motivation.

Another weakness lies in the empirical results.
In Figure 1, the performance gains from the proposed method are marginal in Humanoid-v4, Ant-v4, HalfCheetah-v4, Hopper-v4, and Swimmer-v4.
Furthermore, to substantiate the proposed method, it would be necessary to show that the value error is actually reduced.
Although empirical Q-values are plotted in Figure 2, it would be more informative to plot the difference between the estimated Q-values and the empirical Q-values (i.e., the empirical value error).
The current experimental results do not sufficiently demonstrate the claimed advantages of the proposed approach.

**Questions:**

- If my understanding is correct, the bias in the value function is estimated as the mean of the Bellman error
$\hat{b} = r_i + \gamma Q_1 (s'_i, a'_i) - Q_1 (s_i, a_i)$,
and the target value for $Q_2$ is computed as
$r_i + \gamma Q_2 (s'_i, a'_i) + \hat{b}$.
Please confirm whether this interpretation is accurate or correct me if I am mistaken.

- In lines 236–237, the paper states: "It is additionally noteworthy that  $\epsilon^*_{Q}=\Delta_{Q}$."
Since the definition of $\epsilon^*_Q$ seems missing, please clarify its meaning and provide the mathematical justification for this statement.

- Although Figure 2 plots the empirical Q-values, I suggest plotting the empirical value error instead—approximated as the difference between the estimated and empirical Q-values. Could you provide such a figure to support your claims?

---

### Official Review · Reviewer_NhmY · 2025-10-24

**Soundness:** 1
**Presentation:** 1
**Contribution:** 1
**Rating:** 2
**Confidence:** 3

**Summary:**

The main objective of this work is to estimate and correct the bias present in the Bellman error, which arises because deep reinforcement learning (DRL) algorithms optimize this quantity instead of the true value error. To address this discrepancy, the authors propose a value shaping approach that introduces a bias correction term into the Q-target. Authors evaluate their approach on Mujoco envirnoments, by combining the proposed approach, MSR, with TD3, RRS and TD7.

**Strengths:**

NA

**Weaknesses:**

Major weaknesses:
In general, the paper is quite hard to follow (not due to the theoretical concepts, but due to the writing).
1) Proposition 3.1: This result is already presented in [1] (Theorem 1). You can report this result to improve readability, but you have to specify that this result has already been presentd and proved in [1]. In fact, this cannot be considered as your contribution.
2) Sections 4.1 and 4.2 are quite hard to follow: you introduce b as a result of Proposition 4.3 and you define $\espilon^*_Q$, which is the value error. So, from line 234 to 237, you are just adding notation which might be just confusing. Why eq. 7 is an assumption you make? Equation 6 is the proof that the value error has an additional term.
3) Is there a typo in equation 10? $||f(r)|-|g||\le |f(r)|+|g|$?
4)  The overall discussion in section 4.3 is quite confusing. Once you define $\espilon^*_Q$ in Eq. 7, you can immediately derive $\hat{b}=-\epsilon_Q$, but to derive it you introduce the otimization problem in 11.
5) Since the entire discussion is quite confusing, the derivation of equation (13) is not clear to me. It seems that there is a missing explanation there.
Concerning the results:
A) The results reported in Figure 1 do not show a significant improvement with MRT
B) Depending on the environment/algorithm, the gain might be close to 0, e.g. MRT+TD3 does not improve TD3 in ant-v4.
C) Again, the fact that the Q-values are lower in the early training steps is just (highly) evident in humanoid, halfcheetah, and swimmer, but just for TD7.
D) in 418 "...leading to smaller update magnitudes, which in some cases also results in higher accuracy" is quite vague.
E) In 423: " ...Q-values updates are relatively stable". This depends on the environment and the RL algorithm. In Figure 2, b, c, and d don't show this effect.
F) Same in Figure 3: concerning the Bellman error, the effect of MRT is not the same for all methods and environments.
minor:
1) The name of the method is MRT, where R stands for reward, but then you hightlight that your approach is value shaping rather than reward shaping.
2) 248-249: "we can adjust the parameter of to make". remove "of"
3) 266: "This (eq[8]) transfromed Bellmand error …" Use instead: "The proposed e_Q', the transformed Bellman error, can be …"
[1] Fujimoto, Scott, et al. "Why should i trust you, bellman? the bellman error is a poor replacement for value error." arXiv preprint arXiv:2201.12417 (2022)

**Questions:**

please clarify the weaknesses highlighted

---

### Official Review · Reviewer_d3e2 · 2025-10-26

**Soundness:** 2
**Presentation:** 2
**Contribution:** 1
**Rating:** 2
**Confidence:** 5

**Summary:**

The work investigates a method to reduce the bias in the Bellman error estimation to improve the convergence of policy evaluation methods. The method, called MRT, leverages the fact that an affine translation of the reward function can be implemented without changing the optimal policy of the problem at hand. However, instead of directly learning the affine parameters, the method estimates the Bellman error and subtracts it from the regression target. Experiments are conducted on the MuJoCo benchmark.

While the work focuses on a relevant topic in RL, it does not contain any theoretical contribution, the algorithm is not clearly presented, and the introduced method does not lead to significant improvements. Therefore, the recommendation for this work is "2: reject, not good enough".

**Strengths:**

I. The work focuses on a relevant limitation of value-based methods: the Bellman error is a biased estimate of the value error. Reducing the bias promises improvement of policy evaluation methods.

II. The proposed idea is creative, starting from an interesting perspective.

**Weaknesses:**

A. The presented method is not clearly explained and is misleading at times. This is especially true for section 4.3, which I suggest rewriting entirely.

   i. While the method is motivated from a reward-shaping point of view, the method ends up following the value-shaping paradigm. This shift happens when the bias estimator is set to $- \mathbb{E}[\epsilon_Q]$, on Line 306. First, this result is not justified, and second, it seems that the constraint of the fact that the reward-shaping should be an affine transformation of the original reward function is not taken into account.

   ii. The pseudo-code is unclear as most of the lines are written in text, which increases the ambiguity.

   iii. Equation 10 is wrong. Indeed, $\epsilon'_Q = -1$ and $\epsilon^*_Q = 0$ is a counterexample invalidating Equation 10. Furthermore, it is not justified why minimizing the left term is relevant. Indeed, on Line 260, it is only stated that: "we can minimize the following [Equation 10]".

   iv. The proof of Proposition 4.3 is in the appendix but is not mentioned in the main text.

   v. In Proposition 3.1, $p^{\pi}$ is not defined.

   vi. Line 162, the definition of the modified reward is repeated in Line 165.

   vii. There seems to be an assumption on the reward definition that is not clearly stated in Line 197.

   viii. The presentation of the idea of bias propagation in Remark 4.2 is unclear.

   ix. The term "irrelevant bias" on Line 214 is not defined.

   x. The statement Line 240 is wrong: "We know that the reward can be linearly transformed without affecting the convergence of the optimal policy". Linearly transforming the reward does not change the optimal policy but might affect convergence to it. In fact, it is the primary motivation for transforming the reward.

   xi. Many parentheses are missing around citations, which breaks the ready flow. For example, Line 32, "Bellman (1966)" should be in parentheses.

B. The experimental analysis demonstrates no significant gains with respect to the baseline, and the presentation can be improved.

   i. All curves comparing an algorithm and the presented method are overlapping, which leads to the conclusion that the algorithm is not worth incorporating into existing RL pipelines.

   ii. Figure 2 reports the empirical Q-values. This comparison is not relevant as the proposed algorithm implicitly changes the reward function.

   iii. Similarly to the plot on performance, the plot on Bellman error (Figure 3) does not show any improvement with respect to the baselines.

   iv. The information on the figures is not visible. Increasing the font size and the line width would help.

C. Some related works are missing.

   i. Value shaping for representation learning: Dabney, Will, et al. "The value-improvement path: Towards better representations for reinforcement learning." AAAI 2021.

   ii. Value shaping for minimizing the error propagation: Vincent, Théo, et al. "Iterated Q-Network: Beyond One-Step Bellman Updates in Deep Reinforcement Learning." TMLR 2025.

   iii. Value shaping for convergence acceleration: Farahmand, Amir-Massoud, and Mohammad Ghavamzadeh. "PID accelerated value iteration algorithm." ICLM 2021.

   iv. The work Hao Sun, et al. "Exploit reward shiftingin value-based deep-rl: Optimistic curiosity-based exploration and conservative exploitation vialinear reward shaping." NeurIPS 2022 is cited twice, once in Line 603, and once in Line 608. Please remove one version to avoid double citation.

**Questions:**

N/A

---

### Official Review · Reviewer_Esde · 2025-11-01

**Soundness:** 2
**Presentation:** 2
**Contribution:** 2
**Rating:** 2
**Confidence:** 5

**Summary:**

This paper propose a MRT method to address the estimation error issue. Experiments across Mujoco environment show that MRT improves sample efficiency, reward, and value estimation.

**Strengths:**

1. estimation bias is a fundamenal issue of RL
2. the idea is novel

**Weaknesses:**

1. the theoretical fundation presented in the paper has notable limitations. For Remark 4.1, auther use the exact example as Fujimoto 2022 paper, however this example list an unrealistic example of estimation bias. The estimation bias measures between estimated value and true value over all state and action pairs and it is a stochastic variable. Assuming a fixed additive bias of "+1" across all state-action pairs oversimplifies the nature of estimation bias which only represent the corner case. It fails to capture its statistical properties based on such corner cases to make an significant conclusion of "estimation error is a poor replacement for value error" is problematic. Similar oversimplifications appear in Remark 4.2 and Assumption 4.4, where the propagation and compensation of bias are modeled without accounting for its stochastic properties. These assumptions undermine the rigor of the proposed relationship between TD error and value error. As a result, the connection between Bellman error and value error remains insufficiently justified and requires further theoretical reevaluation.

2. Figure 1, espically for TD3, With MRT, the performance still within the  shaded area which is really hard to see if there exist performance improvment.

3.  Many theoretical parts which is identical to Fujimoto's paper (Proposition 3.1, Remark 4.1, Remark 4.2).

**Questions:**

1. Figure 1-3, the shaded area is confidence interval or std value? please specifiy in the caption.
2. As TD3 state it is focused on overestimation bias but can induce underestimation bias, does this method apply to underestimation bias?

---

### Note · Authors · 2025-12-01

I have read and agree with the venue's withdrawal policy on behalf of myself and my co-authors.